# Small Molecules Inducing Autophagic Degradation of Expanded Polyglutamine Protein through Interaction with Both Mutant ATXN3 and LC3

**DOI:** 10.3390/ijms251910707

**Published:** 2024-10-04

**Authors:** Te-Hsien Lin, Wan-Ling Chen, Shao-Fan Hsu, I-Cheng Chen, Chih-Hsin Lin, Kuo-Hsuan Chang, Yih-Ru Wu, Yi-Ru Chen, Ching-Fa Yao, Wenwei Lin, Guey-Jen Lee-Chen, Chiung-Mei Chen

**Affiliations:** 1Department of Neurology, Chang-Gung Memorial Hospital, Chang-Gung University College of Medicine, Taoyuan 33302, Taiwan; shlin810@cgmh.org.tw (T.-H.L.); windkidll@cgmh.org.tw (W.-L.C.); chlin416@cgmh.org.tw (C.-H.L.); gophy5128@cgmh.org.tw (K.-H.C.); yihruwu@cgmh.org.tw (Y.-R.W.); 2Department of Life Science, National Taiwan Normal University, Taipei 11677, Taiwan; 60943043s@ntnu.edu.tw (S.-F.H.); ichen@ntnu.edu.tw (I.-C.C.); 3Department of Chemistry, National Taiwan Normal University, Taipei 11677, Taiwan; yiruchem@ntnu.edu.tw (Y.-R.C.); cheyaocf@ntnu.edu.tw (C.-F.Y.); wenweilin@ntnu.edu.tw (W.L.)

**Keywords:** ATXN3, autophagosome tethering, LC3, polyglutamine, SCA3 therapeutics, venus-based BiFC

## Abstract

Polyglutamine (polyQ)-mediated spinocerebellar ataxia (SCA), including SCA1, 2, 3, 6, 7, and 17, are caused by mutant genes with expanded CAG repeats, leading to the intracellular accumulation of aggregated proteins, the production of reactive oxygen species, and cell death. Among SCA, SCA3 is caused by a mutation in the ATXN3 (ataxin-3) gene. In a circumstance of polyQ aggregation, the autophagic pathway is induced to degrade the aggregated proteins, thereby suppressing downstream deleterious effects and promoting neuronal survival. In this study, we tested the effects of synthetic indole (NC009-1, -2, -3, -6) and coumarin (LM-022, -031) derivatives as chemical chaperones to assist mutant ATXN3-Q_75_ folding, as well as autophagy inducers to clear aggregated protein. Among the tested compounds, NC009-1, -2, and -6 and LM-031 interfered with *Escherichia coli*-derived ATXN3-Q_75_ aggregation in thioflavin T binding and filter trap assays. In SH-SY5Y cells expressing GFP-fused ATXN3-Q_75_, these compounds displayed aggregation-inhibitory and neurite growth-promoting potentials compared to untreated cells. Furthermore, these compounds activated autophagy by increasing the phosphatidylethanolamine-conjugated LC3 (microtubule associated protein 1 light chain 3)-II:cytosolic LC3-I ratio in these cells. A biochemical co-immunoprecipitation assay by using a mixture of HEK 293T cell lysates containing recombinant ATXN3-Q_75_-Venus-C-terminus (VC) or Venus-N-terminus (VN)-LC3 protein indicated that NC009-1 and -2 and LM-031 served as an autophagosome-tethering compound (ATTEC) to interact with ATXN3-Q_75_ and LC3, and the interaction was further confirmed by bimolecular fluorescence complementation analysis in cells co-expressing both ATXN3-Q_75_-VC and VN-LC3 proteins. The study results suggest the potential of NC009-1 and -2 and LM-031 as an ATTEC in treating SCA3 and, probably, other polyQ diseases.

## 1. Introduction

In polyglutamine (polyQ)-mediated hereditary spinocerebellar ataxia (SCA) types 1, 2, 3, 6, 7, and 17, abnormal expansions of the polyQ stretch in disease-causing proteins trigger the misfolding and aggregation of mutant proteins to interfere with diverse cellular processes [1]. SCA are characterized by cerebellar dysfunction alone or in combination with other neurological abnormalities [2]. Among SCA, SCA3 (also known as Machado–Joseph disease, MJD) is the most common and prevalent form worldwide [3].

SCA3 is caused by a CAG triplet expansion in the ATXN3 (ataxin-3) gene, with 13–36 repeats in healthy individuals and 68–79 repeats in most clinically diagnosed patients [4]. ATXN3 is a deubiquitinating enzyme participating in ubiquitin-dependent proteostasis [5]. Several pathogenic pathways contributing to the neurodegeneration of SCA3 have been shown, which include the dysregulation of transcription, DNA repair dysfunction, impaired ubiquitin–proteasome and autophagy activity, and neuroinflammation [6,7]. The ubiquitin–proteasome system and the autophagy–lysosome pathway are the two major proteolytic mechanisms to degrade abnormal protein aggregates [8]. Previously, we have shown the impaired proteasome function in SCA3, and identified Chinese herbal medicine extracts and their main constituents that can reduce neurotoxicity by enhancing proteasomal activity in SCA3 cellular and induced pluripotent stem cell (iPSC) models [9]. However, when there is an overload of abnormal protein aggregates, the proteasome system alone is not enough to overcome the stress; under this circumstance, the autophagy–lysosome pathway plays an important role in protein quality control.

Autophagy is an essential, conserved self-eating process that cells perform to allow the lysosomal degradation of intracellular protein aggregates. Evidence has shown impaired autophagolysosomal function in SCA3. For example, ATG7 (autophagy-related 7), ATG12 (autophagy-related 12), ATG16L2 (autophagy-related 16 like 2), and LC3 (microtubule-associated protein 1 light chain 3) were significantly increased, whereas BECN1 (beclin-1) levels were decreased in SCA3 post-mortem brains compared to controls [10]. Particularly, ATXN3 binds to and regulates the BECN1 level, whereas mutant ATXN3 also binds to BECN1 but such binding facilitates the degradation of BECN1 by proteasomes, leading to a decreased BECN1 level [11]. The overexpression of BECN1 enhanced the degradation of mutant ATXN3 and ameliorated the disease phenotype and pathology [12,13]. Moreover, P62/SQSTM1 (sequestosome 1), a classical selective autophagy receptor, interacts with polyQ-expanded ATXN3 to regulate aggresomal formation and protect cells against cell death [14]. Therefore, autophagy induction has been implicated as a therapeutic strategy for SCA3 and other polyQ-mediated diseases [15,16]. Although significant progress in the development of therapeutic strategies has been reported, at present, there is no effective treatment for these diseases.

In the past few years, we have focused on screening novel synthetic indole and coumarin derivatives to develop treatment strategies for SCA3. We have reported that indole derivative NC001-8 up-regulated HSPA1A (heat shock protein family A [Hsp70] member 1A) and HSPA8 (heat shock protein family A [Hsp70] member 8) expression and autophagy in SCA3 cells to reduce polyQ aggregation [17]. In addition, indole derivative NC009-1 down-regulated IκBα (nuclear factor kappa B inhibitor alpha)/P65 signaling in SCA3 cells [7] to reduce polyQ aggregation. Moreover, coumarin derivative LM-031 reduced IκBα/P65 and P38 (mitogen-activated protein kinase 14)/STAT1 (signal transducer and activator of transcription 1) signaling in SCA3 cells along with reduced polyQ aggregation [7]. Whether NC009-1 and LM-031 reduce the polyQ aggregate level through direct interfering with mutant polyQ proteins or autophagy induction is not known.

Recently, Li and colleagues have identified indole derivative AN1 and coumarin derivative AN2 as linker compounds for mutant polyQ and LC3 [18] and proposed that the autophagosome-tethering compound (ATTEC) concept could be applied as a therapeutic strategy to enhance the autophagic degradation of mutant polyQ proteins [19]. In this study, we examined whether our in-house indole- and coumarin-derived compounds could interact with both mutant ATXN3 and LC3 to reduce polyQ aggregation and rescue a disease-relevant phenotype in the cell model of SCA3.

## 2. Results

### 2.1. Indole/Coumarin Derivatives and Cytotoxicity

Indole and coumarin derivatives were used to tether mutant polyQ proteins and LC3 together, and the complex was targeted to phagophores for degradation [18]. Based on this, we tested several in-house indole (NC009-1, -2, -3, -6) and coumarin (LM-022, -031) derivatives (Figure 1a) to investigate their potentials as an ATTEC. The cytotoxicity of the test compounds in ATXN3-Q_75_-GFP cells [20] was examined by MTT assay. The results demonstrated the low cytotoxicity of the test compounds in SCA3 SH-SY5Y cells (IC_50_ > 100 μM) (Figure 1b).

### 2.2. Biochemical Test of Anti-PolyQ Aggregation

Trx- and His-tagged C-terminal ATXN3 containing 14 glutamines (Trx-His-ATXN3-Q_14_) or 75 glutamines (Trx-His-ATXN3-Q_75_) [17] were prepared (Figure 2a). When aggregate formation was measured with fluorescence generated by ThT binding after 2 days of incubation at 37 °C, Trx-His-ATXN3-Q_75_ at 2.5–15 µM concentrations aggregated significantly (*p* = 0.045–0.001), whereas Trx-His-ATXN3-Q_14_ aggregation at 2.5–15 µM concentrations was not significant (*p* > 0.05) (Figure 2b). Trx-His-ATXN3-Q_75_ at 2.5 µM was selected to assess the chemical chaperone activity of the test compounds in assisting mutant ATXN3-Q_75_ folding. Significantly increased ATXN3-Q_75_ aggregation was observed after 48 h incubation of ATXN3-Q_75_ protein at 37 °C (from 51% to 100%, *p* < 0.001), which was blocked by congo red, a potent polyQ aggregate inhibitor [21], in a concentration-dependent manner (from 100% to 56–20% in 10–100 µM, *p* < 0.001) (Figure 2c). Among the test compounds, ATXN3-Q_75_ aggregation was significantly reduced by NC009-1, -2, and -6 and LM-031 at 100 µM (from 100% to 79–42%, *p* = 0.003–<0.001) (Figure 2c). Treatments of 100 µM NC009-1, -2, and -6 and LM-031 also led to significant aggregation reductions (from 100% to 60–19%, *p* < 0.001), analyzed using a filter trap assay (Figure 2d)

### 2.3. Neuroprotective Effects on SCA3 ATXN3-Q_75_-GFP SH-SY5Y Cells

Next, we examined the anti-aggregation and neurite outgrowth promotion effects of indole and coumarin compounds on SH-SY5Y cells with inducible ATXN3-Q_75_-GFP expression (Figure 3a). After 7-day retinoic acid-induced neuronal differentiation, ATXN3-Q_75_-GFP-expressing cells demonstrated a significantly increased percentage of aggregated cells in comparison to uninduced cells (8.2% vs. 0.1%, *p* < 0.001) (Figure 3b). The pre-treatment of NC009-1, -2, and -6 and LM-031 at 10 µM led to a reduction in the percentage of aggregated cells from 8.2% to 6.7–5.7% (*p* < 0.001) in ATXN3-Q_75_-GFP-expressing cells (Figure 3b). Treatments of NC009-1, -2, and -6 and LM-031 at 10 µM also led to reduction in insoluble aggregated proteins in cell lysates from 100% to 51–39% (*p* = 0.029–0.004), analyzed using filter trap assay (Figure 3c). Consistent with the effects on ATXN3-Q_75_ aggregation, the NC009-1, -2, and -6 and LM-031 treatments significantly promoted neurite outgrowth in ATXN3-Q_75_-GFP-expressing cells, with a length from 23.2 µm to 26.1–26.9 µm (*p* = 0.002–<0.001), a process from 2.41 to 2.55–2.57 (*p* = 0.004–0.001), and a branch from 1.16 to 1.34–1.37 (*p* = 0.003–<0.001) (Figure 3d). As a positive control [22], trehalose at 10 mM reduced the percentage of aggregated cells (from 8.2% to 7.4%, *p* = 0.044) and insoluble aggregated proteins (from 100% to 45%, *p* = 0.011), and promoted neurite outgrowth (length: 25.5 µm, *p* = 0.014; branch: 1.31, *p* = 0.013) (Figure 3b–d). In summary, NC009-1, -2, and -6 and LM-031 ameliorated the impairments caused by ATXN3-Q_75_ aggregation. As described in [20], no aggregation was seen in retinoic acid-differentiated ATXN3-Q_14_-GFP SH-SY5Y cells.

### 2.4. Autophagic Activation on SCA3 ATXN3-Q_75_-GFP SH-SY5Y Cells

LC3-II is a representative autophagy marker present in autophagic vesicles [23]. A confocal microscopy examination was performed to examine LC3-positive vacuoles (puncta) and ATXN3-Q_75_ aggregates in these cells (Figure 4a). The accumulation of ATXN3-Q_75_-GFP decreased LC3 puncta per cell (from 4.3 to 2.7, *p* < 0.001), and this decrease was improved by trehalose; NC009-1, -2, or -6; or LM-031 treatment (from 2.7 to 4.6–6.7, *p* < 0.001). Meanwhile, the ATXN3-Q_75_ aggregates per cell increased significantly in ATXN3-Q_75_-GFP cells (from 0.00 to 0.31, *p* < 0.001), and the increase was ameliorated by trehalose; NC009-1, -2, or -6; or LM-031 treatment (from 0.31 to 0.12–0.11, *p* < 0.001). The colocalization of ATXN3-Q_75_-GFP with LC3 puncta (from 0.014 to 0.025–0.030, *p* < 0.001) also supported the clearance of ATXN3-Q_75_-GFP in trehalose-; NC009-1-, -2-, or -6-; or LM-031-treated cells by increased autophagy flux. P62 has been known to directly interact with LC3 [24]. The colocalization of LC3 with P62 was significantly increased (from 0.21 to 0.26–0.29, *p* = 0.038–<0.001) (Figure 4b) in trehalose-; NC009-1-, -2-, or -6-; or LM-031-treated cells compared with that found in untreated cells.

Furthermore, the potential of these indole and coumarin compounds to activate autophagy was examined by comparing the ratio of lipid phosphatidyl ethanolamine (PE)-conjugated LC3-II [25] and cytosolic LC3-I between compound-treated and untreated cells (Figure 4c). The induced expression of ATXN3-Q_75_-GFP reduced the ratio of LC3-II:LC3-I (from 100% to 76%), although this reduction was not statistically significant. However, the LC3-II:LC3-I ratio was significantly elevated in cells treated with trehalose; NC009-1, -2, or -6; or LM-031 (from 76% to 102–124%, *p* = 0.043–<0.001). These findings suggest that NC009-1, -2, and -6 and LM-031 enhanced autophagy in SH-SY5Y cells to increase the degradation of ATXN3-Q_75_-GFP.

### 2.5. Interaction of NC009 and LM Compounds with ATXN3-Q_75_ and LC3

To examine if NC009-1, -2, and -6 and LM-031 could serve as an ATTEC through interactions with both mutant ATXN3 and LC3, an assay adapted from the split Venus complementation system [26] was established. The plasmids containing ATXN3-Q_14_ or ATXN3-Q_75_ fused in frame to a Venus-C-terminus (VC) and a Venus-N-terminus (VN) fused in frame to LC3 were constructed (Figure 5a) and expressed transiently in 293T cells. After mixing cell lysates containing ATXN3-Q_75_-VC and cell lysates containing VN-LC3 together, the VN-LC3 in the mixture was co-immunoprecipitated by an anti-ATXN3 or anti-polyQ expansion disease marker antibody (5TF1-1C2) (Figure 5b). In the presence of NC009-1 or -2 or LM-031 (100 µM), the amount of VN-LC3 co-immunoprecipitated by the 5TF1-1C2 antibody was notably increased compared to that without compound treatment (Figure 5c).

In addition, ATXN3-Q_14_- or ATXN3-Q_75_-VC and VN-LC3 proteins were simultaneously expressed in 293T cells from a bidirectional, Tet-responsive promoter for 24 h post-induction of Tet-On 3G *trans*-activator protein with doxycycline (1 µg/mL) (Figure 6a,b). The co-expression of the two genes was confirmed by an immunoblotting of recombinant ATXN3-VC and VN-LC3 proteins (Figure 6c). A microscopic examination of the bimolecular fluorescence complementation (BiFC) of the Venus fluorescence signal generated by the complementation of two non-fluorescent fragments revealed strong Venus fluorescence in 293T cells co-expressing ATXN3-Q_75_-VC and VN-LC3 proteins, whereas a very weak complementary Venus fluorescence was observed in 293T cells co-expressing ATXN3-Q_14_-VC and VN-LC3 proteins (*p* < 0.001) (Figure 6d). The transfected cells were examined for the expression of ATXN3-Q_75_ protein by indirect immunostaining with 5TF1-1C2 antibody, in which the red fluorescence of ATXN3-Q_75_ largely overlapped with the green fluorescence of Venus (ATXN3-Q_75_-VC+VN-LC3) to yield a yellow signal (Figure 6e). Within the examined 24 h period, the expressed ATXN3-Q_75_ was readily seen at the 6 h time point, while green fluorescence from Venus was visible at the 9, 12, and 24 h time points, suggesting that the interaction of ATXN3-Q_75_ with LC3 occurred later after the expression of ATXN3-Q_75_ (Figure 6f).

To test the hypothesis of the test compounds as an ATTEC, 293T cells were transfected with pTRE3G-BI-VN-LC3-ATXN3-Q_75_-VC, and NC009-1, -2, or -6 or LM-031 was added to the transfected cells along with doxycycline; a high-content analysis of overlapped 1C2 (ATXN3-Q_75_) and the Venus signal was performed 24 h post-induction of the *trans*-activator protein (Figure 6b). The overlapping of the green (Venus) and red (ATXN3-Q_75_) signals increased significantly in the transfected cells treated with NC009-1 or -2 or LM-031 (117–119%, *p* = 0.025–0.013) compared to the untreated cells (100%) (Figure 6g). As both the green and red signals are expected in autophagosomes, while the red signal representing ATXN3-Q_75_ disappears in autolysosomes, we also analyzed the green fluorescence of Venus-tagged LC3 puncta as an indicator of autophagy induction. The green fluorescence of Venus-tagged LC3 puncta increased significantly in transfected cells treated with NC009-1 or -2, or LM-031 (121–122%, *p* = 0.048–0.042) compared to untreated cells (100%) (Figure 6g). Together, the results indicate that NC009-1 and -2 and LM-031 could serve as an ATTEC that binds both ATXN3-Q_75_ and LC3 and tethers the mutant proteins to autophagosomes for subsequent autophagic degradation.

## 3. Discussion

The enhancement of autophagy activity has been suggested as a potential therapeutic strategy for neurodegeneration, including polyQ-mediated diseases [16]. In this study, we have shown that synthetic indole (NC009-1, -2, -6) and coumarin (LM-031) derivatives inhibited aggregation and promoted neurite growth in Tet-On ATXN3-Q_75_-GFP SH-SY5Y cells (Figure 3). One of the aggregation-inhibitory mechanisms is the chemical chaperone activity of NC009-1, -2, and -6 and LM-031, which has been shown by the ThT assay (Figure 2). Another mechanism is the increased autophagy activity exerted by these compounds, as supported by the increased ratio of LC3-II:LC3-I in Tet-On ATXN3-Q_75_-GFP SH-SY5Y cells treated with the tested compounds (Figure 4). Furthermore, the co-IP assay using a mixture of 293T cell lysates containing recombinant ATXN3-Q_75_-VC and VN-LC3 proteins demonstrated that NC009-1 and -2 and LM-031 treatment increased the amount of VN-LC3 co-immunoprecipitated by 5TF1-1C2 or by anti-ataxin-3 antibody compared to that without compound treatment (Figure 5). This interaction is further confirmed by BiFC analysis, showing that when pTRE3G-BI-VN-LC3-ATXN3-Q_75_-VC-transfected 293T cells were treated with NC009-1 or -2 or LM-031, the overlapped green (Venus) (indicating the formation of VN-LC3-compound-ATXN3-Q_75_-VC complexes) and red (ATXN3-Q_75_) signals, as well as the green fluorescence of Venus-tagged LC3 puncta, were significantly increased compared to those in cells without compound treatment (Figure 6). Moreover, the cytotoxicity of NC009-1 and -2 and LM-031 is low (> 100 μM; Figure 1) and their BBB penetration ability is predicted (https://www.cbligand.org/BBB/; accessed on 5 October 2021) [27], suggesting their potential for the treatment of neurodegenerative diseases.

Four of the studied compounds, NC009-1, -2, and -6 and LM-031, display chemical chaperone activity through direct binding to ATXN3-Q_75_ to assist folding (Figure 2). These four compounds also display the potential to reduce Aβ42 aggregation [28,29]. As the formation of protein aggregates may be caused by the destabilization of the α-helical structure and the simultaneous formation of a β-sheet [30], the chemical chaperone activity of these compounds may involve the interaction of compounds’ hydrophobic regions with exposed hydrophobic segments of the unfolded protein to protect the protein from aggregation. Since hydrophobic chaperones could promote the conservation of the native structure of proteins, the possibilities of developing chemical chaperones for clinical applications have been explored for various protein-folding diseases [31]. For example, sodium 4-phenylbutyrate (4-PBA) has potential benefits in vitro or in vivo for protein-misfolding diseases such as Huntington’s disease (HD), Parkinson’s disease (PD), and Alzheimer’s disease (AD) [32,33,34]. Both NC009-1 and LM-031 show positive effects in streptozocin-induced hyperglycemic 3× Tg-AD mice [35,36]. Although NC009-1, -2, and -6 and LM-031 have shown chemical chaperone activity on SCA3 in vitro, the in vivo effects remain to be determined.

Several therapeutic strategies, for example, activating AMPK (AMP-activated protein kinase), ULK1 (unc-51-like autophagy-activating kinase 1), BECN1, TFEB (transcription factor EB), or ERRα (estrogen-related receptor alpha), and inhibiting the mTOR (mammalian target of rapamycin), IMPase (inositol monophosphatase), LRRK2 (leucine-rich repeat kinase 2), or c-ABL (Abelson tyrosine kinase), have been developed to enhance autophagy activity in PD models [37]. In addition, some synthetic compounds and clinically used drugs (metformin, sodium valproate, and carbamazepine), as well as natural products (resveratrol, curcumin, trehalose, isorhynchophylline, corynoxine, loganin, and glycyrrhizic acid), have demonstrated their autophagy-enhancing activity through targeting different molecules that are involved in autophagy [37]. Autophagy induction via small molecule compounds has been shown in animal models of SCA3. For example, cordycepin activates autophagy through AMPK phosphorylation to reduce mutant ATXN3 and ameliorate neurological abnormalities in a lentiviral SCA3 mouse model [38]. Carbamazepine also promotes the activation of autophagy and the degradation of mutant ATXN3 in mouse MJD models through activating AMPK [39]. The overexpression of BECN1 activates autophagy to hamper the progression of motor and neuropathological abnormalities of both transgenic and lentiviral SCA3 mice [12]. Calpain inhibitors can improve motor activity and reduce mutant ATXN3 in transgenic zebrafish SCA3 models by increasing LC3-II synthesis [40,41]. Through mTOR inhibition, small-molecule n-butylidenephthalide depletes mutant ATXN3, leading to the alleviation of cerebellar atrophy and improvement in motor function in SCA3 transgenic mice [42].

Pharmacological-targeting autophagy with trehalose has shown promise in the treatment of polyQ diseases. Trehalose, a disaccharide comprising two molecules of glucose, has been reported as an autophagy inducer to speed up the degradation of aggregate-prone proteins, including mutant α-synuclein [43] and polyQ-expanded huntingtin [43] and ATXN3 [44]. Recently, an open-label trial has shown intravenous trehalose administration to 14 patients with SCA3 to be safe and tolerable, along with its effectiveness in alleviating clinical severity scores (ClinicalTrials.gov Identifier: NCT02147886). In addition, a more recent clinical trial of SCA3 patients receiving 100 mg intravenous trehalose showed significant improvements in scores for the assessment and rating of ataxia (SARA) [45]. Due to these positive results, the Food and Drug Administration (FDA) has accepted the Investigational New Drug Application and granted Fast Track designation to SLS-005 (trehalose) for the treatment of SCA, whereas trehalose is still under clinical development. Taken together, these studies have provided significant evidence that activating autophagy is a potential therapeutic strategy for SCA3. Therefore, developing more compounds as autophagy inducers is worthy and of significance.

Recently, ATTEC, another strategy for activating autophagy, has been developed and applied to treat mutant huntingtin-induced neurodegeneration in HD mice and neurons derived from HD-induced pluripotent stem cells [18]. These results show evidence of indole-derivative AN1 and coumarin-derivative AN2 as linker compounds that tether the mutant polyQ to LC3 and lead to subsequent autophagic degradation. Similarly, our study results provide evidence that NC009-1, NC009-2, and LM-031 are able to interact with both mutant ATXN3 and LC3 to enhance the degradation of mutant ATXN3 by autophagy, and support the concept of applying an ATTEC to treat polyQ-mediated diseases or other neurodegenerative diseases caused by impaired protein quality control. However, these findings should be further validated in animal models in the future.

Both indole and coumarin derivatives are widespread heterocycles with diverse biological activities [46,47]. When comparing the two screened coumarin–chalcone hybrids (LM-031 and LM-022) with the reported AN2, both LM-031 and AN2 have a common 5-hydroxy-coumarin core structure, whereas LM-022 bears a 4-hydroxycoumarin core structure. The position of the hydroxy group on the coumarin core structure may be important for an ATTEC, since the LM-022 was non-effective in this study. In addition, both AN1 and the studied NC009 compounds contain a heterocyclic indole core, which in combination with other modifications could be essential for being an ATTEC. Among the NC009 compounds, NC009-1, -2, and -3 have a hydrogen, methyl group, and phenyl group at the C-2 position of indole, whereas NC009-6 has an ethyl group at the C-7 position of indole. The relatively large phenyl and ethyl groups at C-2 of NC009-3 and at C-7 of NC009-6 compared to the smaller hydrogen and methyl group at C-2 of NC009-1 and NC009-2, or a different substituted position (C-7 of NC009-6 and C-2 of NC009-1, -2), may increase steric hindrance to interfere with the linker function. How these compounds interact with ATXN3-Q_75_ and LC3 at the protein structure level should be further studied in the future.

In addition to the autophagy-enhancing effects of NC009-1 and LM-031 shown in this study, our previous studies have revealed the beneficial effects of NC009-1 and LM-031 on the SCA3 model. NC009-1 down-regulates IκBα/P65 signaling and LM-031 reduces IκBα/P65 and P38/STAT1 signaling in SCA3 cells [7]. Therefore, NC009-1 and LM-031 may exert their therapeutic effects by targeting multiple pathways. Dissecting the therapeutic targets of NC009-1 and LM-031 by using animal models should be conducted in the future.

## 4. Materials and Methods

### 4.1. Compounds, Bioavailability, and BBB Permeability Prediction

Indole compounds NC009-1, -2, -3, and -6 and coumarin derivatives LM-022 and -031 were synthesized as described [28,29]. The molecular structure and chemical composition of the synthesized NC009 and LM compounds were characterized by nuclear magnetic resonance spectroscopy.

### 4.2. Cell Culture and Compound Cytotoxicity Assay

The established SH-SY5Y cells expressing GFP-fused C-terminal ATXN3 with 75 glutamines (Flp-In ATXN3-Q_75_-GFP cells) [20] were maintained in Dulbecco’s modified Eagle medium/nutrient mixture F-12 (DMEM/F-12) (Invitrogen, Waltham, MA, USA), supplemented with 10% fetal bovine serum (FBS; Invitrogen), 5 μg/mL blasticidin, and 100 μg/mL hygromycin (InvivoGen, San Diego, CA, USA). Human embryonic kidney 293T cells (ATCC CRL-3216) were cultivated in DMEM containing 10% FBS. Both cell types were cultured at 37 °C under 5% CO_2_ and 95% relative humidity.

For the compound cytotoxicity assay, ATXN3-Q_75_-GFP SY5Y cells (2 × 10^4^) were plated into 96-well dishes, grown for 20 h, and treated with 0.1–100 μM test compounds. After 1 day, 3-(4,5-dimethylthiazol-2-yl)-2,5-diphenyltetrazolium bromide (MTT, 5 mg/mL) (Sigma-Aldrich, St. Louis, MO, USA) was added to the cells and incubated for 3 h at 37 °C. Lysis buffer (10% Triton X-100, 0.1 N HCl, 18% isopropanol) was then added to dissolve the insoluble formazan crystals. The absorbance of the purple-colored solution was measured at optical density (OD) 570 nm by using a FLx800 fluorescence microplate spectrophotometer (Bio-Tek, Winooski, VT, USA). The half-maximal inhibitory concentration (IC_50_) was calculated through the construction of a dose–response curve.

### 4.3. Trx- and His-Tagged ATXN3-Q_14−75_ Proteins and Thioflavin T (ThT) Binding Assay

Plasmids containing ATXN3 C-terminal Q_14−75_ fragments in a prokaryotic expression vector were transformed into *Escherichia coli* BL21(DE3)pLysS (Novagen, Madison, WI 53719, USA), and the expressed Trx-His-ATXN3-Q_14_ and Trx-His-ATXN3-Q_75_ proteins were purified as described [17]. These proteins (0–15 μM) in reaction buffer (150 mM NaCl, 20 mM Tris-HCl pH 8.0) were applied to 96-well black plates and incubated at 37 °C with shaking for 48 h. In addition, the tested NC009 and LM compounds (10–100 µM) were added to Trx-His-ATXN3-Q_75_ protein (2.5 μM) for anti-aggregation test. Congo red (Sigma-Aldrich) was included as a positive control [21]. At the end of incubation, ThT (20 µM final concentration; Sigma-Aldrich) was added to the mixtures and incubated for 20 min at room temperature. The enhanced and red-shifted emission of ThT when binding onto amyloid aggregates was recorded by FLx800 microplate reader (Bio-Tek), with emission/excitation wavelength at 420/485 nm.

### 4.4. Filter Trap Assay

The filter retardation assay was performed to quantify ATXN3-Q_75_ protein aggregates. The purified Trx-His-ATXN3-Q_75_ protein was incubated with test compounds (100 µM) at 37 °C for 2 days as described. In addition, proteins from compound-treated ATXN3-Q_75_-GFP SH-SY5Y cells were prepared (as described in Western blotting). Proteins (0.5 µg of biochemical reaction or 20 µg of cellular lysates) were diluted in 2% SDS in phosphate-buffered saline (PBS) and filtered through a cellulose acetate membrane (0.45-μm pore size; Sterlitech, Auburn, WA, USA) pre-equilibrated in 2% SDS in PBS on a dot-blot filtration apparatus (Bio-Rad Laboratories, Hercules, CA, USA). After washing with 2% SDS buffer, the membrane was blocked in PBS containing 5% skim milk at 4 °C overnight. Insoluble ATXN3-Q_75_ was probed with an anti-His tag (1:2000; OriGene #TA100013, Rockville, MD, USA) or anti-ATXN3 (1:1000; Invitrogen #PA5-26251) antibody. The immune complex on the filter was detected using horseradish peroxidase (HRP)-conjugated goat anti-mouse (#GTX213111-01) or goat anti-rabbit (#GTX213110-01) immunoglobulin G (IgG) antibody (1:5000; GeneTex, Irvine, CA, USA) and a chemiluminescent HRP substrate (Millipore, Billerica, MA, USA). The chemiluminescent signals were captured with an ImageQuant LAS 4000 (GE Healthcare, Chicago, IL, USA) and quantified (Multi Gauge V3.0 software, Fujifilm, Tokyo, Japan).

### 4.5. High-Content Analysis of Aggregation and Neurite Outgrowth Assays

ATXN3-Q_75_-GFP SH-SY5Y cells (5 × 10^4^/well) were plated onto 24-well plates, with retinoic acid addition (10 µM; Sigma-Aldrich) on day 1 to initiate neuronal differentiation. On day 2, the cells were pretreated with disaccharide trehalose (10 mM, as a positive control) [22] or the studied compounds (10 μM) for 8 h, followed by the addition of doxycycline (5 μg/mL, Sigma-Aldrich) to induce ATXN3-Q_75_-GFP expression. The cells were kept in the medium containing retinoic acid, the test compound, and doxycycline for 6 days, with the culture medium changed on day 5. ATXN3-Q_75_ aggregation and neurite outgrowth assays were performed on day 8 using a high-content imaging system (ImageXpress Micro Confocal; Molecular Devices).

For the aggregation assay, cells were stained with Hoechst 33342 (0.1 µg/mL; Sigma-Aldrich) at 37 °C for 30 min. Images of the cells were automatically obtained at excitation/emission wavelengths of 482/536 nm (FITC [fluorescein isothiocyanate] filter) for enhanced GFP and 377/447 nm (DAPI [4′,6-diamidino-2-phenylindole] filter) for nucleus counting. ATXN3-Q_75_ aggregation was determined by Transfluor technology [48] based on GFP fluorescence intensity. For the neurite outgrowth assay, cells were fixed with 4% paraformaldehyde for 30 min, permeabilized with 0.1% Triton X-100 for 10 min, and blocked by non-specific binding with 3% bovine serum albumin for 20 min. The primary TUBB3 (neuronal class III β-tubulin) antibody (1:1000; BioLegend #802001, San Diego, CA, USA) was used to stain cells at 4 °C overnight, followed by Alexa Fluor 555-donkey anti-rabbit IgG secondary antibody (1:1000; Invitrogen #A-31572) staining for 2 h at room temperature. Nuclei were detected using DAPI (0.1 μg/mL; Sigma-Aldrich). Cell images were obtained at excitation/emission wavelengths of 531/593 nm (TRITC [tetramethylrhodamine] filter) for stained neurites and 377/447 nm (DAPI filter) for nuclei. To quantify neurite outgrowth, microscopic images were segmented with multi-colored masks to assign each outgrowth to a cell body for quantification. Neurite total length (μm) and numbers of process (primary neurites originating from neuronal cell body) and branch (secondary neurites extended from primary neurites) were analyzed (Neurite Outgrowth Application Module; Molecular Devices). In general, 104 cells in each biological replicate were analyzed.

### 4.6. Western Blotting

For the immunoblotting analysis, ATXN3-Q_75_-GFP SH-SY5Y cells were seeded in 6-well plates (3 × 10^5^/well) and treated with retinoic acid, test compounds, and doxycycline, as described. On day 8, cells were collected and lysed in buffer (50 mM Tris-HCl pH 8.0, 150 mM NaCl, 2 mM EDTA pH 8.0, 0.1% SDS, 0.5% sodium deoxycholate, 1% NP-40) containing the protease inhibitor cocktail (Sigma-Aldrich). After sonication, the lysates were centrifuged (12,000× *g* for 10 min at 4 °C) and protein concentrations determined (Bradford protein assay; Bio-Rad, Hercules, CA, USA). Aliquots of protein (20 µg) were separated by electrophoresis using 12% SDS-polyacrylamide gel followed by transfer to a polyvinylidene fluoride membrane. After being blocked with 5% skim milk at 4 °C overnight, the membrane was stained with LC3 (1:1000; Medical & Biological Laboratories Co. #PM036, Tokyo, Japan) or GAPDH (glyceraldehyde 3-phosphate dehydrogenase) (1:1000, MDBio #30000002, Taipei, Taiwan) primary antibody at 4 °C overnight. The immune complexes were detected using HRP-conjugated goat anti-rabbit IgG antibody and chemiluminescent substrate, as described.

### 4.7. Immunocytochemical Staining and Confocal Microscope Examination of ATXN3-Q_75_ Cells

ATXN3-Q_75_-GFP SY5Y cells were seeded onto 12-well plates (1 × 10^5^/well) containing poly-L-lysine-coated coverslip on day 1; treated with retinoic acid, test compounds, and doxycycline on day 2; and fixed, permeabilized, and blocked on day 8, as described. The primary anti-LC3 antibody (1:1000; Cell Signaling #83506, Danvers, MA, USA) or anti-P62 antibody (1:1500; Cell Signaling #5114) was used to stain cells at 4 °C overnight. After washing with PBS containing 0.1% Tween 20 (PBST), cells were incubated for 2 h at room temperature with Cy5-conjugated (1:1000, Jackson ImmunoResearch 715-175-150) or Alexa Fluor 555-conjugated (1:500; Invitrogen #A-31572) secondary antibody, and washed three times with PBST. Nuclei were detected using DAPI (0.1 μg/mL; Sigma-Aldrich). The stained cells on coverslips were mounted onto microscope slides with Vectashield antifade mounting medium (Vector Laboratories, Newark, CA, USA) and examined using a Zeiss LSM880 confocal microscope (Carl Zeiss AG, Oberkochen, Baden-Württemberg, Germany). Cell images were acquired at excitation/emission wavelengths of 649/666 nm, 482/536 nm, 555/582 nm, and 385/461 nm for stained LC3 puncta, aggregates, P62, and nuclei, respectively. For colocalization, 300 cells from three independent experiments were analyzed and quantified for the proportion of pixels that LC3 (red) colocalized with P62 (yellow) using Mander’s overlap coefficient (MOC) in ZEN software version 2.1 (Carl Zeiss AG).

### 4.8. pcDNA3-ATXN3-Q_14−75_-VC and pcDNA5-VN-LC3 Constructs and Expression

*Eco*RI-*Nco*I ATXN3 C-terminal Q_14_ and Q_75_-containing fragment (amino acids 251–361 in full-length ATXN3-Q_14_ [NM_004993]; [20]) and *Nco*I-*Not*I Venus C-terminal fragment (VC_212-239_) [49] were subcloned into *Eco*RI- and *Not*I-digested pcDNA3 vector (Invitrogen) to generate pcDNA3-ATXN3-Q_14−75_-VC plasmid for producing ATXN3-Q_14−75_-VC fusion protein linked by a 6-amino acid polypeptide linker (containing *NcoI* restriction site). In addition, *Bam*HI-*Xba*I Venus-N-terminal fragment (VN_1-211_) [49] and *Xba*I-*Bam*HI LC3 cDNA fragment (MAP1LC3B, NM_022818) were subcloned into *Bam*HI-digested pcDNA5/FRT/TO vector (Invitrogen) to generate pcDNA5-VN-LC3 plasmid for producing VN-LC3 fusion protein linked by a 2-amino acid dipeptide linker (containing *Xba*I restriction site). The recombinant pcDNA3-ATXN3-Q_14_-VC, pcDNA3-ATXN3-Q_75_-VC, or pcDNA5-VN-LC3 plasmid (2 μg) was transiently transfected into 293T cells (2 × 10^5^/6-well) using T-Pro non-liposome transfection reagent II (T-Pro Biotechnology #JT97-N002M, New Taipei City, Taiwan) according to the supplier’s instruction. After 48 h, cell lysate containing ATXN3-Q_14_-VC, ATXN3-Q_75_-VC, or VN-LC3 fusion protein was prepared using cytoskeleton (CSK) buffer containing 10 mM PIPES, 100 mM NaCl, 300 mM sucrose, 1 mM EGTA, 1 mM MgCl_2_, and 1 mM DTT and quantified, as described.

### 4.9. Co-Immunoprecipitation Assay

In this case, 293T cell lysates with transiently expressed ATXN3-Q_14_-VC, ATXN3-Q_75_-VC, or VN-LC3, and mixture of ATXN3-Q_75_-VC and VN-LC3 (100 μg of each) in 400 μL CSK buffer were incubated at 4 °C under gentle rotation. To test the concept of the ATTEC, 100 μM of the compound (NC009-1, -2, and -6 or LM-031) was added to the lysates containing ATXN3-Q_75_-VC and VN-LC3 proteins. After overnight incubation, ATXN3-Q_14−75_-VC protein was immunoprecipitated with rabbit ataxin-3 (1:1000; Invitrogen #PA5-26251) or mouse 5TF1-1C2 antibody (1:500; Merck #MAB1574, Darmstadt, Germany) at 4 °C overnight. In addition, normal (non-specific) mouse IgG antibody (1:500; Jackson ImmunoResearch #015-000-003, West Grove, PA, USA) was added to the lysates containing ATXN3-Q_75_-VC and VN-LC3 proteins for comparison. The protein–antibody mixtures were added to protein G Mag Sepharose magnetic beads (Cytiva #28951379, Marlborough, MA, USA) and incubated at room temperature for 20 min under gentle agitation. After removing the supernatants, the bead–protein–antibody mixtures were washed with CSK buffer and the immunoprecipitated proteins were eluted in SDS sample loading buffer (50 mM Tris pH 6.8, 2% SDS, 10% glycerol, 1% β-mercaptoethanol, 12.5 mM EDTA, 0.02% bromophenol blue) by heating at 50 °C for 10 min. The input and eluted proteins were separated on a 10% SDS–polyacrylamide gel and immunoblotted with LC3 (1:2000; MBL International #PM036, Woburn, MA, USA), GFP (1:1000; Bioman #egfp001r, New Taipei City, Taiwan), ATXN3 (1:1000; Invitrogen #PA5-26251), or expanded polyQ (5TF1-1C2, 1:1000; Merck #MAB1574) antibody at 4 °C overnight. The immune complexes were detected using HRP-conjugated goat anti-rabbit (#GTX213110-01) or goat anti-mouse (#GTX213111-01) IgG antibody (1:5000; GeneTex) and chemiluminescent substrate, as described.

### 4.10. pTRE3G-BI-VN-LC3-ATXN3-Q_14−75_-VC Construct and Expression

pTRE3G-BI-VN-LC3-ATXN3-Q_14−75_-VC was constructed by subcloning the ATXN3-Q_14−75_-VC *Kpn*I-*Xba*I fragment and VN-LC3 *Bam*HI fragment of the aforementioned plasmids into the corresponding restriction sites in multiple cloning site (MCS)-2 (ATXN3-Q_14−75_-VC fragment) or MCS-1 (VN-LC3 fragment) of the pTRE3G-BI vector (Clontech, Mountain View, CA, USA). The resulting plasmid allows the simultaneous expression of ATXN3-Q_14−75_-VC and VN-LC3 proteins from a bidirectional, Tet-responsive promoter that binds the Tet-On 3G *trans*-activator protein in the presence of doxycycline and co-transfected pCMV-Tet3G. Next, 293T cells (2 × 10^5^) were plated onto 6-well plates on day 1. Upon the next day, the constructed pTRE3G-BI-VN-LC3-ATXN3-Q _14−75_-VC (2 μg) together with pCMV-Tet3G (0.5 μg) were transiently co-transfected into cells using GenJet in vitro DNA transfection reagent (SignaGen Laboratories #SL100488, Rockville, MD, USA). Doxycycline (1 μg/mL) was added to cells to induce ATXN3-Q_14−75_-VC and VN-LC3 expression on day 3. Cell lysates were prepared at 24 h post-induction and analyzed by Western blot using ataxin-3 (1:1000; Invitrogen #PA5-26251) or LC3 (1:2000; MBL International #PM036) antibody. In addition, at 0, 6, 9, 12, and 24 h post-transfection, the pTRE3G-BI-VN-LC3-ATXN3-Q_75_-VC transfected cells were fixed, permeabilized, blocked, and stained with 5TF1-1C2 (1:1000; Merck #MAB1574) antibody at 4 °C overnight, followed by Cy5-donkey anti-mouse IgG secondary antibody (1:1000; Jackson ImmunoResearch #715-175-151) staining for 2 h at room temperature. Nuclei were counterstained with DAPI (0.1 μg/mL; Sigma-Aldrich). Cell images were obtained at excitation/emission wavelengths of 482/536 nm (FITC filter) for Venus, 631/692 nm (Cy5 filter) for stained expanded ATXN3-Q_75_ protein, and 377/447 nm (DAPI filter) for detecting nuclei (ImageXpress Micro Confocal; Molecular Devices).

### 4.11. High-Content Aggregation Analysis of Transfected 293T Cells

On day 3, doxycycline (1 μg/mL) was added to the aforementioned transfected 293T cells along with NC009-1, -2, or -6 or LM-031 (10 μM). At 24 h post-transfection, the transfected cells were fixed, permeabilized, blocked, and stained with 5TF1-1C2 antibody, as described. After nucleus staining (DAPI, 0.1 μg/mL), the captured images were analyzed as described for overlapped ATXN3-Q_75_ and Venus signals and the green fluorescence of Venus-tagged LC3 puncta using the high-content imaging system, as described.

### 4.12. Statistical Analysis

The presented data were expressed as mean ± SD of three independent experiments. To compare the differences between groups, one-way ANOVA (analysis of variance) was performed, with Tukey’s post hoc test where appropriate. All *p* values were two-tailed, with values lower than 0.05 considered as being statistically significant.

## 5. Conclusions

We have shown that the indole and coumarin derivatives NC009-1, NC009-2, and LM-031 interfered with ATXN3-Q_75_ protein aggregation in ThT binding and filter trap assays. These three compounds displayed good aggregation-inhibitory and neurite growth-promoting potentials in Tet-On ATXN3-Q_75_-GFP SH-SY5Y cells. These compounds activated autophagy by increasing the LC3-II:LC3-I ratio in ATXN3-Q_75_-expressing cells. A biochemical co-IP assay indicated that NC009-1, NC009-2, and LM-031 served as an ATTEC that interacted with ATXN3-Q_75_ and LC3, and the interaction was further confirmed by BiFC analysis in 293T cells co-expressing both ATXN3-Q_75_-VC and VN-LC3 proteins (Figure 7). The study results suggest the potential of NC009-1, NC009-2, and LM-031 in treating SCA3 and, probably, other polyQ diseases.

## Figures and Tables

**Figure 1 ijms-25-10707-f001:**
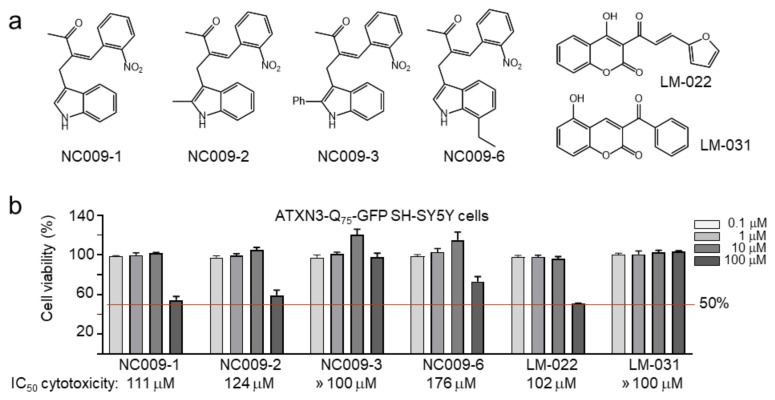
Studied indole and coumarin compounds. (**a**) Structure and formula of indole (NC009-1, -2, -3, -6) and coumarin (LM-022, -031) derivatives. (**b**) Cytotoxicity of test compounds against human ATXN3-Q_75_-GFP SH-SY5Y cells determined by MTT assay. Cells were treated with each test compound (0.1–100 μM) and cell viability was analyzed the next day (*n* = 3). For normalization, the relative viability of untreated cells was set at 100%. Shown below are the IC_50_ cytotoxicity values.

**Figure 2 ijms-25-10707-f002:**
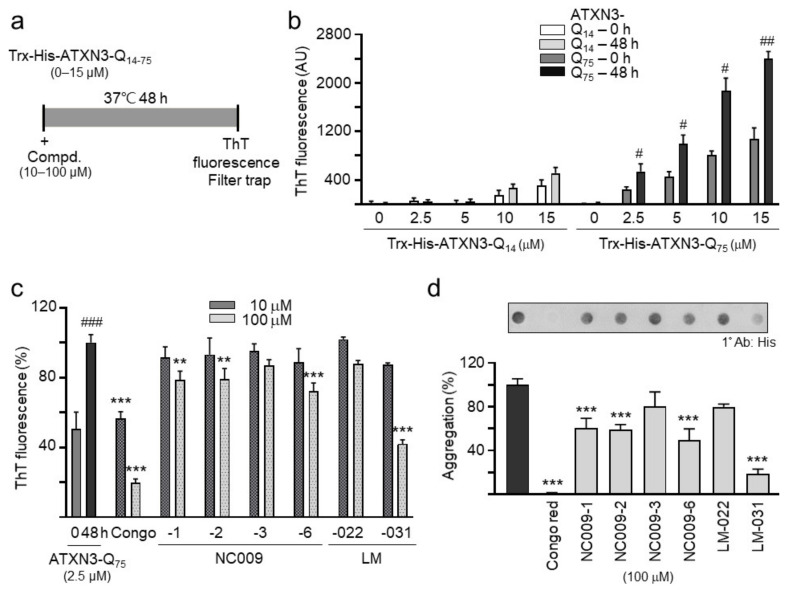
Biochemical ATXN3-Q_14−75_ aggregation assays. (**a**) Experimental flow chart. Trx-His-ATXN3-Q_14–75_ proteins (0–15 µM) were incubated at 37 °C for 48 h; aggregation was monitored by ThT fluorescence assay. In addition, Trx-His-ATXN3-Q_75_ protein (2.5 µM) was incubated with congo red (as a positive control) or tested compounds (10–100 μM) at 37 °C for 48 h, and aggregation was monitored by ThT fluorescence and filter trap assays. (**b**) ThT binding assay for ATXN3 aggregation. Trx-His-ATXN3-Q_14–75_ proteins (0–15 µM) were incubated at 37 °C for 48 h, and aggregation was monitored by measuring ThT fluorescence intensity (*n* = 3). (**c**) The effects of the studied compounds (10–100 μM) on the aggregation of Trx-His-ATXN3-Q_75_ protein (2.5 µM) (*n* = 3) was analyzed by ThT binding assay. To normalize, the relative ThT fluorescence of Trx-His-ATXN3-Q_75_ protein alone after 48 h incubation was set as 100%. (**d**) Filter trap assay of studied compounds (100 μM) (*n* = 3). Insoluble ATXN3-Q_75_ retained on cellulose acetate membrane was stained with anti-His antibody. The relative insoluble ATXN3-Q_75_ without compound addition was set as 100%. *p* values: 0 h vs. 48 h incubation (#: *p* < 0.05, ##: *p* < 0.01, ###: *p* < 0.001), or with vs. without compound addition (**: *p* < 0.01, ***: *p* < 0.001).

**Figure 3 ijms-25-10707-f003:**
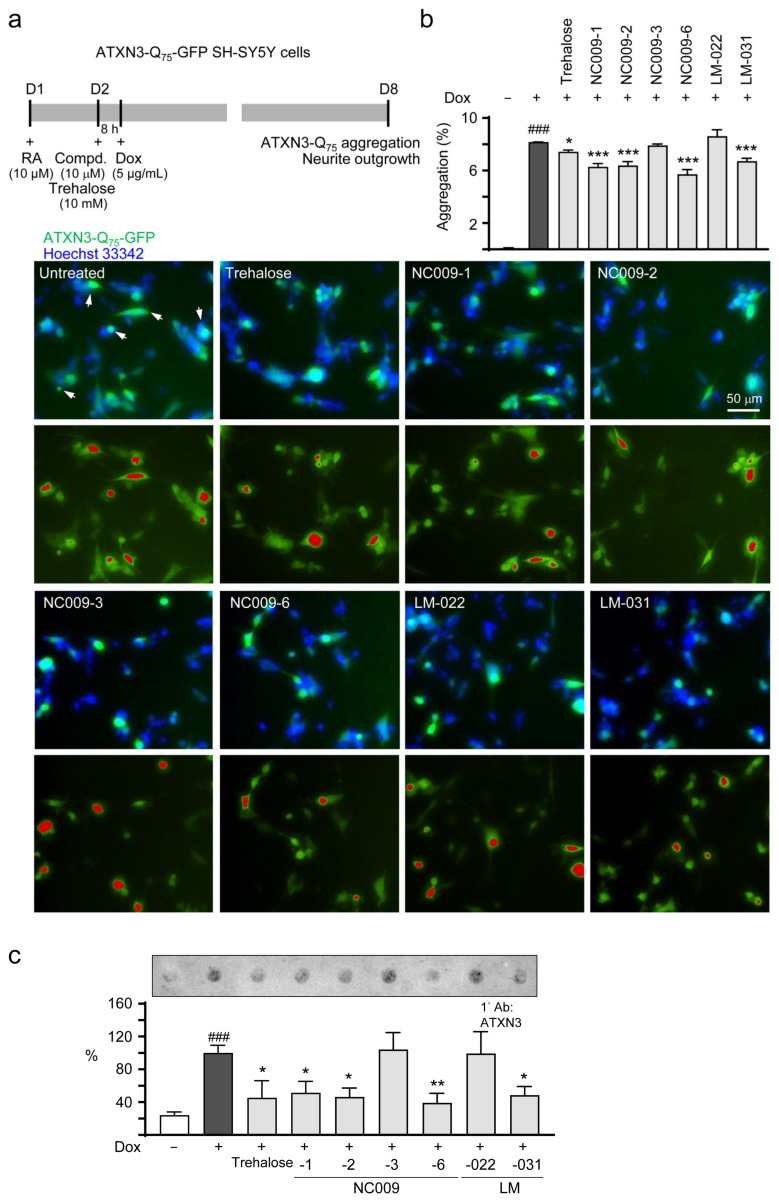
ATXN3-Q_75_ aggregation and neurite outgrowth assessments in SH-SY5Y cells expressing ATXN3-Q_75_-GFP. (**a**) Experimental flow chart. On day 1, cells were plated in the presence of retinoic acid (RA, 10 μM) to initiate neuronal differentiation. On day 2, studied compounds (10 μM) or trehalose (10 mM; as a positive control) were added to the cells for 8 h, followed by the induction of ATXN3-Q_75_-GFP expression with doxycycline (Dox, 5 µg/mL). On day 8, ATXN3-Q_75_ aggregation and neurite outgrowth were assessed. (**b**) Images and analysis of ATXN3-Q_75_ aggregation (percentage of aggregated cells) without or with compound treatment (*n* = 3). Nuclei were counterstained with Hoechst 33342 (blue). The white arrows in the untreated photo indicate aggregates. Images with red-colored masks to assign aggregates for quantification are shown below. (**c**) Filter trap assay of ATXN3-Q_75_ aggregation (*n* = 3). Insoluble ATXN3-Q_75_ retained on the cellulose acetate membrane was visualized by immunostaining with anti-ATXN3 antibody. The relative insoluble ATXN3-Q_75_ without compound treatment was set as 100%. (**d**) High-content neurite outgrowth images (TUBB3 stain, orange) and analysis of ATXN3-Q_75_-GFP cells without or with compound treatment (*n* = 3). Nuclei were counterstained with DAPI (blue). Images with multi-colored masks to assign the outgrowth of a cell body for quantification are shown below. The arrows in the uninduced photo indicate process (red) and branch (white). The length, process, and branch of neurite were calculated. *p* values: with vs. without doxycycline induction (##: *p* < 0.01, ###: *p* < 0.001), or with vs. without compound treatment (*: *p* < 0.05, **: *p* < 0.01, ***: *p* < 0.001).

**Figure 4 ijms-25-10707-f004:**
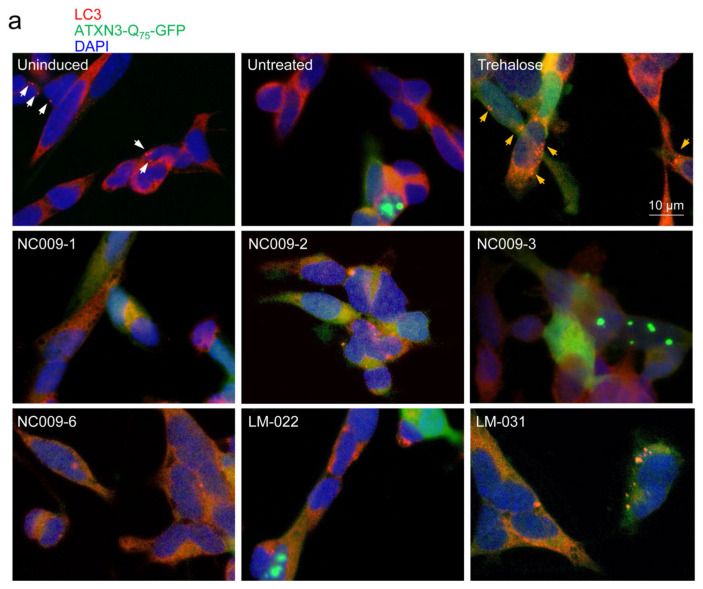
Autophagy activation to scavenge protein aggregates in ATXN3-Q_75_-GFP SH-SY5Y cells. Cells were treated with trehalose (10 mM) or studied compounds (10 μM), as described in Figure 3. On day 8, LC3 puncta, ATXN3-Q_75_ aggregates, and LC3-II:LC3-I ratio were assessed. (**a**) Confocal microscopy images of LC3-positive puncta and ATXN3-Q_75_ aggregates in ATXN3-Q_75_-GFP SH-SY5Y cells treated with trehalose or studied compounds (green, expressed ATXN3-Q_75_-GFP protein; red, LC3; blue, DAPI-stained nuclei). The arrows (white) in the uninduced photo indicate puncta. The arrows (orange) in the trehalose-treated photo indicate overlapped puncta and aggregates. Shown below are analyses of LC3 puncta and ATXN3 aggregates per cell, and colocalization coefficient of ATXN3-Q_75_-GFP with LC3 puncta. (**b**) Immunofluorescence staining of LC3 and P62 in ATXN3-Q_75_-GFP SH-SY5Y cells treated with trehalose or studied compounds (red, LC3; yellow, P62; blue, DAPI-stained nuclei). The arrows (orange) in the trehalose-treated photo indicate overlapped puncta and P62. Shown below is colocalization coefficient of LC3 with P62. (**c**) Relative LC3-I and LC3-II levels analyzed by Western blot using LC3 and GAPDH (as a loading control) antibodies. Relative LC3-II:LC3-I ratio in uninduced cells was set at 100% (*n* = 3). *p* values: with vs. without doxycycline induction (###: *p* < 0.001), or with vs. without compound treatment (*: *p* < 0.05, **: *p* < 0.01, ***: *p* < 0.001).

**Figure 5 ijms-25-10707-f005:**
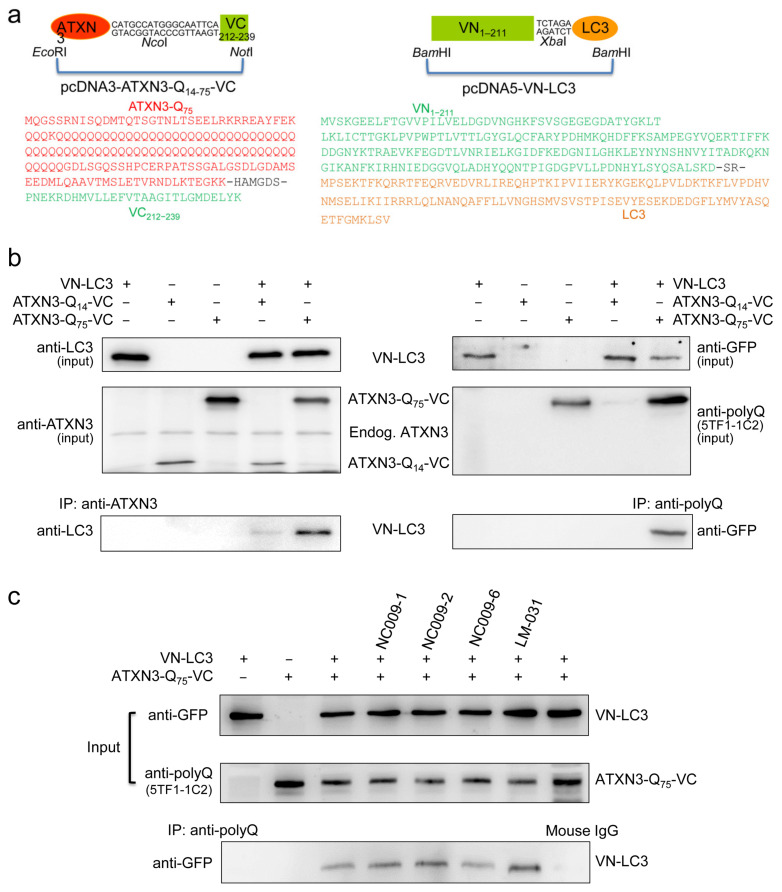
Interaction of studied compounds with ATXN3-Q_75_ and LC3. (**a**) pcDNA3-TXN3-Q_14−75_-VC and pcDNA5-VN-LC3 constructs and amino acid sequences of the expressed recombinant proteins. To produce ATXN3-Q_14−75_-VC, a 105-amino acid ATXN3-Q_14_ or a 166-amino acid ATXN3-Q_75_-containing *Eco*RI-*Nco*I DNA fragments (marked in red: ATXN3-Q_75_) was linked to a 28-amino acid VC (C-terminal fragment of Venus)-containing *Nco*I-*Not*I DNA fragment (marked in green) through a 6-amino acid linker (containing *Nco*I restriction site). To produce VN-LC3, a 211-amino acid VN (N terminal fragment of Venus)-containing *Bam*HI-*Xba*I DNA fragment (marked in green) was linked to a 125-amino acid LC3-containing *Xba*I-*Bam*HI DNA fragment (marked in orange) through a 2-amino acid linker (containing *Xba*I restriction site). (**b**) Co-immunoprecipitation (Co-IP) of ATXN3-Q_14−75_-VC and VN-LC3: 293T cell lysates containing 100 μg of ATXN3-Q_14_-VC, ATXN3-Q_75_-VC, or VN-LC3, and mixtures of ATXN3-Q_14−75_-VC and VN-LC3 were subjected to co-IP. The expression of ATXN3-Q_14−75_-VC or VN-LC3 fusion proteins in cell lysates were confirmed by anti-ATXN3 (detecting ATXN3-Q_14−75_-VC protein), anti-polyQ (5TF1-1C2, detecting only ATXN3-Q_75_-VC protein), or anti-LC3 and anti-GFP (detecting VN-LC3 protein) antibody staining of protein blot. (**c**) The 100 μg individual (ATXN3-Q_75_-VC or VN-LC3) or combined cell lysates in the presence or absence of the test compound (100 μM) were immunoprecipitated with 5TF1-1C2 antibody, and the presence of VN-LC3 protein was detected by immunoblot analysis with anti-GFP antibody. Mouse IgG antibody was used as a negative control in the co-IP experiment. The experiments were repeated three times with similar results.

**Figure 6 ijms-25-10707-f006:**
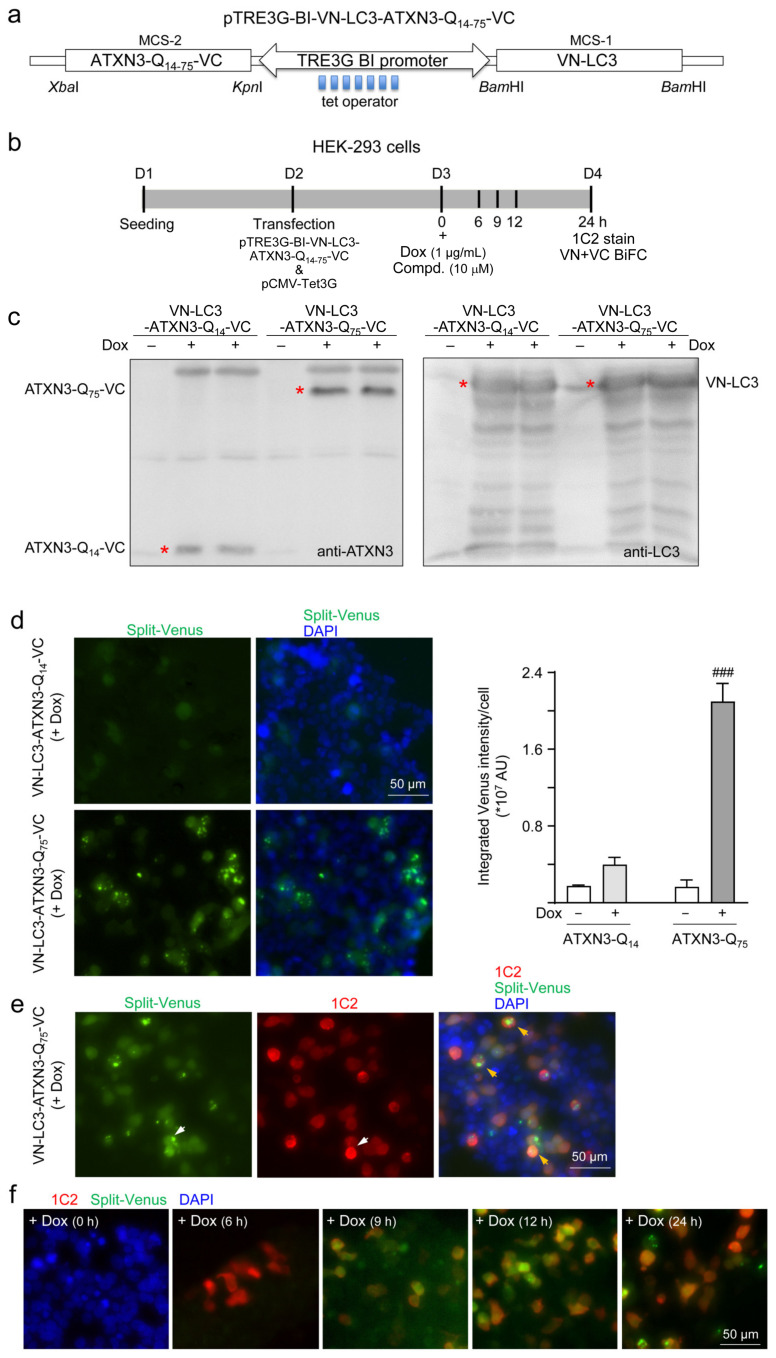
Autophagic degradation of expanded polyQ protein in 293T cells with dual expression of VN-LC3 and ATXN3-Q_75_-VC proteins. (**a**) pTRE3G-BI-VN-LC3-ATXN3-Q_14−75_-VC construct for doxycycline-inducible dual expression of VN-LC3 and ATXN3-Q_14−75_-VC. The pTRE3G-BI promoter consists of 7 repeats of a 19 bp tetracycline (tet) operator sequence. VN-LC3 was cloned in *Bam*HI site of multiple cloning site (MCS)-1; ATXN3-Q_14−75_-VC was cloned between *Xba*I and *Kpn*I sites of MCS-2. (**b**) Experimental flow chart. On day 1, 293T cells were plated, and they were co-transfected with pTRE3G-BI-VN-LC3-ATXN3-Q_14−75_-VC and pCMV-Tet3G (4:1 ratio) on day 2. Doxycycline (1 μg/mL) alone or along with test compound (10 μM) was added to cells to induce trans-activator protein and hence ATXN3-Q_14−75_-VC and VN-LC3 expression on day 3. At 24 h post-induction, cell lysates were prepared for immunoblot analysis of VN-LC3 and ATXN3-Q_14−75_-VC protein expression. In addition, high-content analysis of overlapped 1C2 (ATXN3-Q_75_)/Venus signal was performed at 6, 9, 12, and 24 h post-induction. (**c**) Immunoblot analysis of ATXN3-Q_14−75_-VC (anti-ATXN3 antibody) and VN-LC3 (anti-LC3 antibody) at 24 h post-induction. The red asterisk indicates antibody-stained ATXN3-Q_75_-VC, VN-LC3, or ATXN3-Q_14_-VC proteins. (**d**–**f**) High-content images (Venus–BiFC, green; ATXN3-Q_75_-1C2 stain, red) of 293 transfected cells at 24 h (**d**,**e**) or 6, 9, 12, and 24 h post-induction of protein expression (f). Nuclei were counterstained with DAPI (blue). The arrows in (**e**) indicate split-Venus (white arrow), 1C2-stained ATXN3-Q_75_ aggregates (white arrow), and overlapped split-Venus and ATXN3-Q_75_ aggregates (orange arrow). (**g**) High-content images and analyses of IC2 and Venus (left) and Venus-tagged LC3 puncta (right) in compound-treated transfected cells at 24 h post-induction of protein expression. Nuclei were counterstained with DAPI (blue). The arrows (orange) in the induced photo indicate overlapped IC2 and split-Venus. *p* values: with vs. without doxycycline induction (###: *p* < 0.001), or with vs. without compound treatment (*: *p* < 0.05).

**Figure 7 ijms-25-10707-f007:**
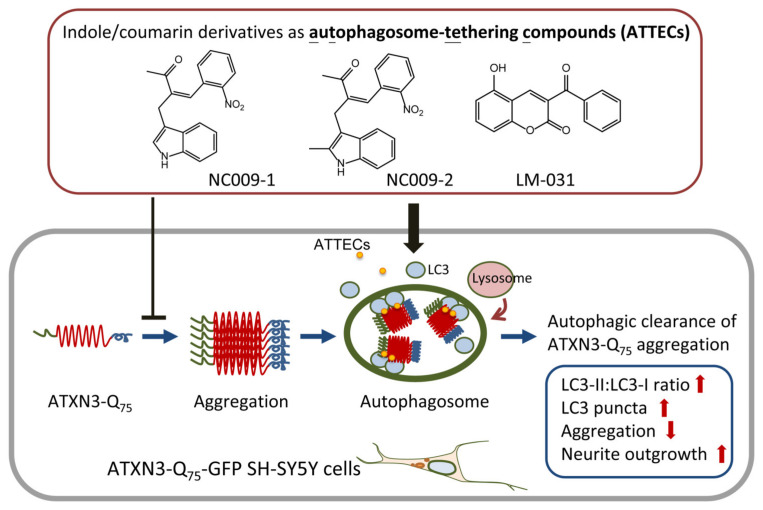
Graphical summary of indole/coumarin derivatives NC009-1, NC009-2, and LM-031 as ATTECs to induce autophagic degradation of expanded polyQ protein in ATXN3-Q_75_ SH-SY5Y cells.

## Data Availability

The raw data that support the findings of this study are available from the corresponding authors on reasonable request.

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
