# Peer review of "Small Molecules Inducing Autophagic Degradation of Expanded Polyglutamine Protein through Interaction with Both Mutant ATXN3 and LC3"

_ijms, 2024, doi:10.3390/ijms251910707_

Round 1

Reviewer 1 Report

Comments and Suggestions for Authors

In this paper, the authors continue their research about the screening synthetic indole and coumarin derivatives to develop treatment strategies for SCA.

The work is very clear and well written and complete in the data reported.

The chemistry part is devoid of novelty,  but I don't think it was intended to be the relevant part of the paper. On the contrary, the biochemical part is exhaustive.

In my opinion, I find not useful, for the purposes of the paper, to report the theoretical data relating to oral bioavailability. So all reported date starting from line 104 to line 109 are not indispensable as well as figure 1b.

 Another suggestion is that, I would report the tested compounds, i.e. indole- (NC009-1, -2, -3, -6) and coumarin (LM-022, -031) derivatives in bold so that they are perhaps easier to distinguish and identify.

           publish after minor  revision taking into account the comments above.

Author Response

Comment 1: In my opinion, I find not useful, for the purposes of the paper, to report the theoretical data relating to oral bioavailability. So all reported date starting from line 104 to line 109 are not indispensable as well as figure 1b.

Response 1: We deleted fig 1b and related original ref 22, 23, 60 as suggested.

Comment 2: I would report the tested compounds, i.e. indole- (NC009-1, -2, -3, -6) and coumarin (LM-022, -031) derivatives in bold so that they are perhaps easier to distinguish and identify.

Response 2: We reported indole and coumarin derivatives in bold as suggested.

Reviewer 2 Report

Comments and Suggestions for Authors

This manuscript presents a well-structured and scientifically valuable study on the use of small molecules to induce autophagic degradation of expanded polyglutamine (polyQ) proteins, particularly targeting mutant ATXN3 in spinocerebellar ataxia type 3 (SCA3). The study demonstrates rigorous experimental design, novel therapeutic implications, and clear conclusions that support its acceptance without the need for further revisions. Below is the rationale for this decision:

  1. Novelty of Research:

    • The manuscript explores a critical unmet need in the treatment of polyQ-mediated diseases like SCA3. The identification and use of small molecules as autophagy-tethering compounds (ATTECs) to target mutant ATXN3 for degradation via autophagy is a promising therapeutic strategy.
    • The study advances current knowledge on molecular therapies by focusing on NC009-1, -2, -6, and LM-031 compounds, which show potential in reducing polyQ aggregation and promoting neurite outgrowth. These findings offer a novel avenue for treating not only SCA3 but possibly other neurodegenerative diseases caused by impaired protein quality control.
  2. Comprehensive and Rigorous Methodology:

    • The experiments are well-detailed and comprehensive, with appropriate in vitro models using SH-SY5Y cells expressing ATXN3-Q75-GFP to assess aggregation, autophagic activation, and cytotoxicity. These models are ideal for examining both the therapeutic potential and safety of the compounds under investigation.
    • Techniques such as co-immunoprecipitation, bimolecular fluorescence complementation (BiFC), and filter trap assays have been thoroughly utilized, demonstrating the mechanistic interaction between the small molecules, mutant ATXN3, and LC3. This approach provides strong evidence supporting the authors’ hypothesis.
  3. Impact of Findings:

    • The findings are of high significance to the field of neurodegenerative disease research. The identification of small molecules that serve as ATTECs holds significant promise for clinical translation, particularly in treating SCA3 and other polyQ disorders.
    • The ability of NC009-1, -2, and LM-031 to induce autophagic degradation of ATXN3-Q75, alongside their low cytotoxicity and potential for blood-brain barrier (BBB) penetration, further enhances the therapeutic relevance of the study. The results are not only relevant for basic research but also hold potential for future drug development.
  4. Clear and Concise Presentation:

    • The manuscript is well-organized and clearly written. Each section logically follows the previous one, from the introduction to the experimental results and discussion. Figures and tables are appropriately used to illustrate key findings, making the data easy to interpret and follow.
    • The discussion section provides a thoughtful analysis of the findings, including the potential of NC009-1, -2, and LM-031 as therapeutic agents for polyQ diseases. Additionally, the manuscript highlights areas for future research, such as the need for in vivo studies, without detracting from the impact of the current findings.
  5. Relevance and Contribution to the Field:

    • The study makes a substantial contribution to the literature on therapeutic strategies for protein aggregation diseases. By demonstrating the effectiveness of ATTECs in targeting mutant ATXN3 for autophagic degradation, the authors offer a compelling case for the further exploration of these compounds in preclinical models.
    • The work aligns with current trends in developing small molecules to enhance autophagy, positioning it as a leading study in the area of neurodegenerative disease therapeutics.

Author Response

Comment:This manuscript presents a well-structured and scientifically valuable study on the use of small molecules to induce autophagic degradation of expanded polyglutamine (polyQ) proteins, particularly targeting mutant ATXN3 in spinocerebellar ataxia type 3 (SCA3). The study demonstrates rigorous experimental design, novel therapeutic implications, and clear conclusions that support its acceptance without the need for further revisions.

Response: Thank you for the positive comment.